# The Role of Insect Symbiotic Bacteria in Metabolizing Phytochemicals and Agrochemicals

**DOI:** 10.3390/insects13070583

**Published:** 2022-06-26

**Authors:** Man Zhao, Xingyu Lin, Xianru Guo

**Affiliations:** Henan International Laboratory for Green Pest Control, College of Plant Protection, Henan Agricultural University, Zhengzhou 450002, China; zhaoman821@henau.edu.cn (M.Z.); xingyulin666666@163.com (X.L.)

**Keywords:** insect microbiota, plant secondary substance, insecticide resistance, detoxifying enzymes, insect immune system

## Abstract

**Simple Summary:**

To counter plant chemical defenses and exposure to agrochemicals, herbivorous insects have developed several adaptive strategies to guard against the ingested detrimental substances, including enhancing detoxifying enzyme activities, avoidance behavior, amino acid mutation of target sites, and lower penetration through a thicker cuticle. Insect microbiota play important roles in many aspects of insect biology and physiology. To better understand the role of insect symbiotic bacteria in metabolizing these detrimental substances, we summarize the research progress on the function of insect bacteria in metabolizing phytochemicals and agrochemicals, and describe their future potential application in pest management and protection of beneficial insects.

**Abstract:**

The diversity and high adaptability of insects are heavily associated with their symbiotic microbes, which include bacteria, fungi, viruses, protozoa, and archaea. These microbes play important roles in many aspects of the biology and physiology of insects, such as helping the host insects with food digestion, nutrition absorption, strengthening immunity and confronting plant defenses. To maintain normal development and population reproduction, herbivorous insects have developed strategies to detoxify the substances to which they may be exposed in the living habitat, such as the detoxifying enzymes carboxylesterase, glutathione-S-transferases (GSTs), and cytochrome P450 monooxygenases (CYP450s). Additionally, insect symbiotic bacteria can act as an important factor to modulate the adaptability of insects to the exposed detrimental substances. This review summarizes the current research progress on the role of insect symbiotic bacteria in metabolizing phytochemicals and agrochemicals (insecticides and herbicides). Given the importance of insect microbiota, more functional symbiotic bacteria that modulate the adaptability of insects to the detrimental substances to which they are exposed should be identified, and the underlying mechanisms should also be further studied, facilitating the development of microbial-resource-based pest control approaches or protective methods for beneficial insects.

## 1. Introduction

Insects, which are the most abundant and widely distributed species in the animal kingdom, can survive and reproduce under various conditions [1,2]. The diversity and adaptability of insects are closely related to their symbiotic microbes, including bacteria, fungi, viruses, protozoa and archaea [3]. In insects, these microbes inhabit the exoskeleton, gut, blood cavity, salivary gland, and other organs, as well as individual cells, accounting for 1–10% of insect biomass and playing critical roles in many aspects of the biology and physiology of insects [4,5,6,7].

During the interaction between microbial symbionts and insects, the insects provide the habitat and nutrition for microbes, and in return, these symbionts help the host insects with food digestion, nutrition absorption, defense responses to pathogens, and xenobiotic metabolism, while also promoting insect development and reproduction [8,9]. For example, the fungal-yeast-like symbiotes in planthoppers and aphids are vital for the synthesis of essential amino acids and for maintaining the vitamin supply in the insect host [10,11,12]. The polydnavirus from parasitoid wasps can disorder the immune system of host insects to ensure the survival of wasp offspring, and three partiti-like viruses identified from the African armyworm (*Spodoptera exempta*) can enhance the resistance of *S. exempta* to nucleopolyhedrovirus [13,14]. For wood-feeding lower termites, they rely on symbiotic flagellates to decompose the lignocelluloses in their plant diet, and methanogenic archaea to produce methane [15,16]. In terms of symbiotic bacteria, these comprise the most abundant microorganism species in insects and are mainly distributes in the gut, including Proteobacteria, Bacteroidetes, Firmicutes, Clostridia, Actinomycetes, and others, and contribute to the development, behavior, communication, and adaptation of host insects [3,17,18,19,20,21,22]. In addition, the composition of bacterial communities in insects can be influenced by food resources, environmental factors, pathogenic microbials, or the detrimental substances to which they are exposed [23,24,25].

Early studies on the symbiotic microbes of insects have mainly relied on traditional isolation and culture methods, but a major limitation of these methods is that many microbes are uncultured, and their functional roles cannot be studied in vivo. In recent years, the rapid development of high-throughput metagenomic sequencing technology and methods for rearing germ-free insects has promoted research on functional characterization of the microorganisms in insects, especially the symbiotic bacteria [26]. For some insect species linked to agriculture (such as pests, pollinators, and parasitic enemies), their development, learning behavior, and resistance evolution are highly relevant to gut bacteria [4,24,27]. For insect vectors transmitting human diseases (such as mosquitoes), some symbiotic bacteria, influencing the vector transmitting efficiency or reproduction of mosquitoes, can be targets for potential public disease control [28,29]. In past decades, extensive studies have been conducted on insect bacterial community diversity and interactions of bacteria with host insects (Figure 1). This review focuses on the research progress of insect symbiotic bacteria in metabolizing phytochemicals and agrochemicals (insecticides and herbicides), which are two main kinds of substances insects encounter in their life histories. Finally, the microbe-based pest control approaches, pest resistance management strategies, and protective methods of natural enemy insects that may apply in the future should be examined.

## 2. Insect Bacteria Confer Resistance to Phytochemicals

In nature, more than half of insects are herbivores, which damage different kinds of crops and even cause economical losses [30,31]. To defend themselves from attack by insect herbivory, plants have evolved various defensive mechanisms, including production of phytochemicals such as alkaloids, terpenoids, phenols, and some other secondary substances that show detrimental effects on the growth and survival of insects or attract the natural enemies of herbivores [32]. To cope, herbivorous insects have developed several strategies to detoxify the ingested phytochemicals, including the concerned biochemical counteradaptations [33].

In addition to biochemical responses, insect symbiotic bacteria play key roles in countering plant defenses [7,34]. Before feeding, the oral secretions of some insect herbivores contain a few effector molecules that suppress the antiherbivore defenses, and some bacteria (belonging to the genera *Stenotrophomonas*, *Pseudomonas*, and *Enterobacter*), identified from oral secretions of *Leptinotarsa decemlineata*, are also responsible for plant defense suppression [35,36,37]. After ingestion, the consumed plant tissue enters the digestive tract of the insects, and the gut bacterial community is able to help hosts with food digestion, nutrition absorption, and countering the toxic or harmful phytochemicals from the plant diet [9,34]. For generalist insects, to some extent, their polyphagous habits rely on several symbiotic bacteria to adapt to phytochemicals from different host plants [38]. For example, when fed with *Arabidopsis thaliana*, the gut bacteria of *Trichoplusia ni* were dominated by *Shinella*, *Terribacillus* and *Propionibacterium*, which are known to have the ability to degrade the plant allelochemical glucosinolate; when feeding on *S**olanum lycopersicum*, the relative abundances of *Agrobacterium* and *Rhizobium* able to degrade alkaloids were significantly increased [39]. However, specialist insects may need specific bacteria to degrade the toxic compounds in their host plants, such as *Enterococcus* sp. from *Hyles euphorbiae* and *Brithys crini**,* which have the ability to tolerate alkaloid and latex [40].

Terpenes are a class of toxic phytochemicals that are highly present in coniferous plants. To overcome these toxic compounds, the pests that colonize these plants metabolize the toxic compounds with the aid of symbiotic bacteria. For example, the gut bacteria *Serritia*, *Pseudomonas*, and *Rahnella* from *Dendroctonus ponderosae* have a strong ability to degrade monoterpenes and diterpene acids because these genera contain the majority of the genes that participate in terpene degradation [41,42]. For another mountain pine beetle, *Dendroctonus valens*, its gallery lengths and body weight were significantly suppressed when fed on a diet containing α-pinene at 6 and 12 mg/mL, and three bacterial strains (*Serratia* sp., *Pseudomonas* sp., and *Rahnella aquatilis*) degraded 20–50% of α-pinene [43]. However, the role of these bacteria in degrading terpene has not been verified in the two pine beetles in vivo. A further study on the gut microbiota of the pine weevil (*Hylobius abietis*) found that the weevil can degrade substantial amounts of diterpene in its plant diet, and this degradability was significantly reduced after eliminating gut microbes with antibiotics and then restored again after supplying a normal gut microbial community. When inoculating the gut bacterial community with dehydroabietic acid for five days, the amount of bacteria significantly reduced, and the metagenomic analysis results showed that beetles fed on Norway spruce contained 10 degradation genes (*dit*), which were almost eliminated after treating with antibiotics [44]. In another weevil (*Curculio chinensis*), the gut bacteria from the genus *Acinetobacter* degraded tea saponin and used it as source of carbon and nitrogen [45]. Moreover, some insects, such as *Rhodnius prolixus*, counteract the toxic effects of azadirachtin (a triterpenoid compound of terpenes) by promoting the gene of equivalent NF-kB transcription factor (*RpDorsal*) and antimicrobial peptide (*defC AMP*), as well as the abundance of the gut bacterium *Serratia marcescens* [46].

Alkaloids, a kind of plant phytochemical, are neurotoxic to a wide range of insects, and most of them have been used as botanical agrochemicals for pest control [47]. Although alkaloids exhibit toxicity to most insects, a few species still show high tolerance to these substances, such as *Hypothenemus hampei*, which can consume coffee beans rich in the alkaloid caffeine. Later researchers found that the tolerance of *H*. *hampei* to caffeine is underpinned by its gut microbiota. After eliminating the gut microorganisms with antibiotics, the fitness of *H*. *hampei,* fed on a caffeine-treated diet, declined and showed no decrease in caffeine concentration in their frass. Through a culture-dependent approach, a gut bacterium, *Pseudomonas fulva,* was isolated, which processed a gene coding one subunit of a caffeine demethylase, and the reinstatement of *P**. fulva* in germ-free *H*. *hampei* recovered its capacity to degrade caffeine [48]. As another important phytochemical, phenols inhibit herbivorous insects by inducing reactive oxygen production. When feeding on unripe olives, the olive fly *Bactrocera olea* requires the gut bacterium *Erwinia dacicola* to overcome the toxic phenolic glycoside in unripe olives [49]. Metagenomic analysis revealed that the bacterium *Novosphingobium* sp. in *D**. valens* possesses putative genes involved in the degradation of naringenin, and the survival rate of *D. valens* on a naringenin-treated diet significantly increased when supplied with *Novosphingobium* sp. [50]. In addition, the gut bacteria *Acinetobacter* sp. in *Lymantria dispar* also use condensed tannins as a carbon source [51]. In the cabbage stem flea beetle *Psylliodes chrysocephala*, when its gut bacteria were removed with antibiotics, the beetles accumulated 11.3-fold higher levels of unmetabolized isothiocyanates compared to control beetles, and the isothiocyanate degradation ability was restored when reintroducing the bacteria *Pantoea* sp. Pc8 in antibiotic-fed beetles [52]. For the phytochemical oxalate, the gut bacterium *Ishikawaella capsulata* in stinkbug *Megacopta punctatissima* encodes genes for oxalate decarboxylase, suggesting the possible role of the bacterium in oxalate detoxification [53]. In human, calcium oxalate is formed if the food-derived oxalate cannot be metabolized, which can result in kidney stone disease, so the identification of insect bacteria able to degrade oxalate may act as a novel treatment for kidney stone patients [54] (Table 1).

Apart from detoxification roles, some insect bacteria can convert phytochemicals into pheromone compounds and thus influence the chemical communication of host insects [56]. For instance, the gut bacteria *Pantoea agglomerans*, *Klebsiella pneumonia,* and *Enterobacter cloacae* of *Schistocerca gregaria* can use the plant-derived vanillic acid to produce guaiacol and phenol, which are two main components of the locust cohesion pheromone [20]. In the mine beetle *Chrysolina herbacea*, its gut bacteria has the ability to metabolize terpenoids into pheromone compounds [57]. In addition to phytochemicals, the *Bacillus* species isolated from the male rectum of *Bactrocera dorsalis* can directly produce sex pheromone components (2,3,5-trimethylpyrazine and 2,3,5,6-tetramethylpyrazine) by using glucose and threonine as the substrates. After treating male flies with antibiotics, the levels of the two components were significantly reduced [58]. These findings suggest that some insect bacteria may be an ideal choice for microbe-based pest control because their products can disorder the normal aggregation or mating behavior of pests.

## 3. Association between Gut Bacteria and Insects’ Adaptation to Agrochemicals

To promote crop yield and quality, many agrochemicals are applied on fields to control the dominant economic pests, but frequent application of these chemicals has also resulted in severe health and environmental issues, as well as the resistance evolution of pests to these widely used chemicals, and nontarget toxicity to natural enemies or pollinators [59,60,61,62,63]. To find alternatives with novel modes of action against pests, genetically modified (GM) crops that express insecticidal proteins derived from *Bacillus thuringiensis* (Bt) have been developed and commercially planted since 1996, but resistant populations of target pests were also recorded after several years [64,65,66].

Amino acid mutation of target sites and upregulation of detoxification enzymes or transporters mainly confer the resistance evolution of insects to these agrochemicals [67,68,69], but recently, insect-associated bacteria have also been reported to directly or indirectly participate in the adaptability of insects to agrochemicals (Table 2).

### 3.1. Symbionts Directly Degrade Agrochemicals

When exposed to agrochemicals, the survival, development, behavior, as well as the composition and abundance of gut bacteria in target insects are affected [70]. However, under long-term high selection pressure of agrochemicals, the target insects also evolve resistance to the exposed agrochemicals, and, in some cases, the diversity and abundance of gut microbiota between resistant insect populations and susceptible insect populations are significantly different [71,72,73]. Compared with susceptible insect strains, the uniquely enriched gut bacteria in resistant insects should receive more attention, because these bacteria may participate in conferring insect resistance to some agrochemicals [74]. In *Aedes albopictus*, an important urban pest that can transmit viruses such as dengue, Zika, and chikungunya, the 16S rRNA sequencing results of intestinal bacteria between deltamethrin-resistant and -sensitive strains showed that the bacteria *Serratia oryzae* and *Acinetobacter junii* had higher abundance in resistance strains, and these strains may help *Ae. albopictus* develop resistance to deltamethrin, but their roles have not been verified in vitro or in vivo [72]. In deltamethrin-resistant *Spodoptera frugiperda*, the isolated bacterium *Arthrobacter nicotinovorans* grew better in the selective media and cleared 54.9% of deltamethrin [75]. Similarly, the gut symbionts *Burkholderia* from *Riptortus pedestris* and *Cletus punctiger* metabolize fenitrothion (an organophosphorus agrochemical) into nontoxic substances and use them as the available carbon source, thus promoting the development of host insects and conferring their resistance to fenitrothion. These bacteria are also present in the soil, and when treating field soil with fenitrothion for one month, the bacterial community increased to 10^7^ to 10^8^ CFU/g, of which >80% showed fenitrothion-degrading activities, suggesting that the insects may acquire fenitrothion-degrading bacteria from the soil [76,77]. Furthermore, in *Blatta orientalis*, the degradation rates of bacteria *Pseudomonas aeruginosa* G1, *Stenotrophomonas maltophilia* G2, and *Acinetobacter lwoffii* G5 to α-endosulfan were all >80%, which may facilitate insecticide resistance evolution and make cockroaches difficult to control [78]. In *Anopheles gambiae*, the gut bacteria *Sphingobacterium*, *Lysinibacillus*, *Streptococcus,* and *Rubrobacter* are highly associated with its resistance to permethrin [79]. Apart from insecticides, the insect gut bacterium *Acetobacter tropicalis,* isolated from *Drosophila melanogaster,* is also responsible for atrazine detoxification (one herbicide), and the restoration of *A**. tropicalis* in germ-free flies reduces atrazine toxicity. Genome sequencing results showed that this bacterium contains candidate genes *atzA*, *atzB*, and *atzC*, which are involved in atrazine metabolism [80]. Furthermore, the gut bacteria *Serratia marcescens* and *Pseudomonas protegens* in *Nasonia vitripennis* also confer atrazine resistance. When exposed to atrazine for several generations, the bacterial densities of *S*. *marcescens* and *P*. *protegens* in *N*. *vitripennis* significantly increased. The degradation rates of these strains to atrazine were 20% and 10%, respectively, and whole-genome sequencing results also indicated the possession of the atrazine metabolism genes [24].

During the interaction of insect gut microbes with agrochemicals, some detoxification enzymes, encoded by the genes of symbionts, also play important roles in the metabolism of agrochemicals. The results of comparative genomics analysis showed that the gut symbiont *Citrobacter* sp. of *Bactrocera dorsalis* encodes genes of phosphatase hydrolase, and the gene expression levels are higher when exposed to trichlorphon. When antibiotic-treated flies were supplied with *Citrobacter* sp., the hosts obtained insecticide resistance to trichlorphon [81]. The bacterial esterase and carboxylesterase facilitated the degradation of indoxacarb in *Plutella xylostella* [82]. The above findings suggest that the degradation effects of insect gut bacteria directly mediate insect resistance to agrochemicals.

### 3.2. Indirect Regulation of Insect Resistance by Gut Bacteria

In addition to direct degradation, insect microbes can regulate insect resistance to agrochemicals by activating the detoxification the enzyme or immune system in hosts [83,84]. For instance, after treatment with polymyxin B, the survival rate of *Bombyx mori* exposed to chlorpyrifos was significantly lower, and 16S rRNA gene sequencing results showed that the abundances of the genera *Stenotrophomonas* and *Enterococcus* were decreased. When supplying germ-free silkworms with *S. maltophilia*, the host resistance to chlorpyrifos was enhanced. However, this bacterium cannot directly degrade chlorpyrifos in the gut, but by promoting the activity levels of acetylcholinesterase in hosts [85]. In *Culex pipiens*, the abundance of the intestinal bacterium *Aeromonas hydrophila* in deltamethrin-resistant populations was found to be much higher. After eliminating the gut bacteria of the resistant strains with antibiotics, its resistance level was reduced by 66%, while the enzyme activity of cytochrome P450 monooxygenases (CYP450s) in the hosts was reduced by 58%. Supplying *A**. hydrophila* restored the resistance and enzyme activity of CYP450s, indicating that *A**. hydrophila* increases the resistance of hosts to deltamethrin by enhancing the activity of CYP450s [86]. In addition, the *Enterococcus* sp. isolated from the guts of *P**lutella xylostella* enhance insecticide resistance to chlorpyrifos by regulating the expression of an antimicrobial peptide named gloverin [87]. After exposure to imidacloprid, the abundance of *Wolbachia* in *Nilaparvata lugens* increased, and removing this bacterium reduced the enzyme activity of CYP450s, while the transcript level of NlCYP4CE1 also significantly decreased. This result suggested that *Wolbachia* enhances the resistance of hosts to imidacloprid by promoting the expression of NlCYP4CE1 [88]. For pollinators such as the honeybee (*Apis mellifera*), the gut microbiota promotes the expression of some immune-related genes (hymenoptaecin, defensin1) and detoxification-related genes (*CYP450s*, *GST,* and *catalase*), and thus increase honeybee tolerance to thiacloprid, tau-fluvalinate, or flumethrin [89,90].

**Table 2 insects-13-00583-t002:** Symbiont-mediated insect resistance to agrochemicals.

Bacteria and Insect Host	Target Agrochemical	Description	Reference
*Serratia oryzae* and *Acinetobacter junii* in *Aedes albopictus*	Deltamethrin	*S. oryzae* and *A. junii* had higher abundance in deltamethrin-resistant strain (by 16S rRNA sequencing)	[72]
*Arthrobacter nicotinovorans* in *Spodoptera frugiperda*		Cleared 54.9% of deltamethrin (by LC-MS)	[75]
*Burkholderia* strains in *Riptortus pedestris* and *Cletus punctiger*	Fenitrothion	Bacteria metabolized fenitrothion into nontoxic substance, and insects infected with fenitrothion-degrading *Burkholderia* strains had higher survival rate and larger body size (by HPLC).	[76,77]
*Pseudomonas aeruginosa* G1, *Stenotrophomonas maltophilia* G2, and *Acinetobacter lwoffii* G5 in *Blatta orientalis*	α-endosulfan	Degradation rates of *P. aeruginosa* G1, *S. maltophilia* G2, and *A. lwoffii* G5 to α-endosulfan were 88.5%, 85.5%, and 80.2%, respectively (by HPLC)	[78]
*Sphingobacterium*, and *Lysinibacillus Streptococcus* and *Rubrobacter* in *Anopheles gambiae*	Pyrethroid	*Sphingobacterium*, *Lysinibacillus*, *Streptococcus,* and *Rubrobacter* significantly more abundant in resistant mosquitoes (by 16S rRNA gene sequencing)	[79]
*Acetobacter tropicalis* in *Drosophila melanogaster*	Atrazine	Atrazine exposure reduced relative abundance of *Acetobacter*, and restoration of *A. tropicalis* in germ-free flies reduced atrazine toxicity bacterium contained genes involved in atrazine metabolism (by 16S rRNA gene sequencing)	[80]
*Serratia marcescens* and *Pseudomonas protegens* in *Nasonia vitripennis*		Bacterial densities of *S. marcescens* and *P. protegens* in atrazine-fed *N. vitripennis* significantly increased, and degradation rates to atrazine were 20% and 10%, respectively; both contained genes involved in atrazine metabolism (by 16S rRNA gene sequencing, HPLC, whole-genome sequencing)	[24]
*Stenotrophomonas maltophilia* in *Bombyx mori*	Chlorpyrifos	Enhanced host resistance to chlorpyrifos by increasing activities of acetylcholinesterase (by 16S rRNA gene sequencing, qRT-PCR, GC-MS)	[85]
*Aeromonas hydrophila* in *Culex pipiens*	Deltamethrin	Increased the resistance of hosts to deltamethrin by enhancing activities of CYP450s (measurement of activity levels of enzyme)	[86]
*Enterococcus* sp. in *Plutella xylostella*	Chlorpyrifos	Enhanced insecticide resistance to chlorpyrifos by regulating expression of antimicrobial peptide named gloverin (by using a UV spectrophotometer at 293 nm absorbance and qRT-PCR)	[87]
*Wolbachia* in *Nilaparvata lugens*	Imidacloprid	Enhanced resistance of hosts to imidacloprid by promoting expression of NlCYP4CE1 (by 16S rRNA gene sequencing, qRT-PCR, measurement of activity levels of enzyme)	[88]
gut bacteria in *Apis mellifera*	Thiacloprid, tau-fluvalinate and flumethrin	E=Enhanced insecticide resistance of hosts by promoting expression of immune-related genes and detoxification-related genes (by 16S rRNA gene sequencing, qRT-PCR, HPLC)	[89,90]

## 4. Degradation of Other Detrimental Substances by Insect Bacteria

As the main secondary metabolites produced by mycotoxigenic fungi, mycotoxins have been found in nearly all agricultural goods, and they can cause severe human health problems and economic losses during livestock production [91]. To prevent the contamination of agricultural commodities by mycotoxins, many strategies have been recommended; there has recently been increasing interest in detoxification methods involving functional microbes isolated from natural samples [92,93,94]. Under natural conditions, some herbivorous insects co-occur with mycotoxigenic fungi [95]. Accordingly, they must be able to tolerate exposure to these mycotoxins to ensure that they normally develop and reproduce. Thus, they may be useful sources of functional microbes capable of detoxifying mycotoxins. To date, most of the reported mycotoxin-degrading microorganisms were isolated from noninsect systems (such as soil, water, or contaminated crops), with only one study demonstrating that *Symbiotaphrina kochii*, which is a symbiont in the tobacco beetle *Lasioderma serricorne*, can detoxify mycotoxins, including deoxynivalenol, ochratoxin A, and sterigmatocystin [96]. Future studies should identify and isolate additional functional microbes in insects that are highly tolerant to mycotoxins [97].

The overuse and abuse of antibiotics in livestock production and the treatment of human disease have resulted in severe problems associated with antibiotic resistance and antibiotic residues [98]. The gut microbes of *Musca domestica* and *Hermetia illucens* can efficiently degrade oxytetracycline (>54.5%), implying that insect gut microorganisms may be useful for eliminating antibiotic residues [99,100,101]. Some insect bacteria can produce antimicrobial compounds that contribute to protection from pathogens. For example, the gut bacterium *Enterococcus mundtii* in *Spodoptera littoralis* can secrete an antimicrobial (mundticin KS) against the invading bacteria, and the purified mundticin can cure larvae infected with *E. faecalis* [21]. Furthermore, cockroaches also carry bacteria that can produce metabolites or proteins with potential industrial applications, such as the antibiotic-producing *Streptomyces* strain, *Bacillus* strain, *Enterococcus* strain, and *Pseudomonas* species, all of which may be suitable for development as pharmaceuticals or plant protection products and provide opportunities for biotechnological application [102].

## 5. Conclusions and Future Perspectives

Insect microbiota are critical for metabolizing diverse detrimental substances. Future research on beneficial insects, including pollinators and natural enemies of pests, should consider the utility of microorganisms as biocontrol agents that can provide protection from the effects of toxic substances. Regarding pests, the role of their microbial partners should be monitored when developing new strategies for controlling pests or decreasing the vector competence of pests (e.g., the death of male insects and parthenogenesis caused by *Wolbachia* and *Rickettsia* species), but this may require genetic modifications. Furthermore, identifying microbes in insects able to detoxify harmful compounds may have important implications for bioremediation or for limiting the toxicity of xenobiotics.

## Figures and Tables

**Figure 1 insects-13-00583-f001:**
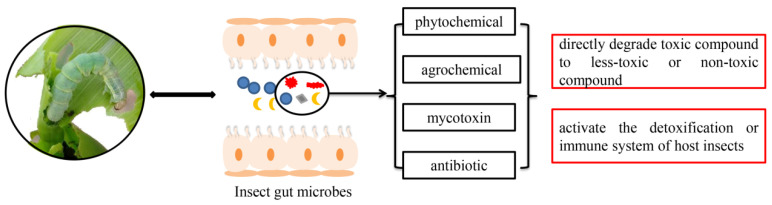
An overview of symbiont-mediated detoxification in insects.

**Table 1 insects-13-00583-t001:** Symbiont-mediated detoxification of phytochemicals.

Plant Allelochemical		Functional Bacteria and Host	Description	Reference
Terpenoid	Monoterpene	*Serritia marcescens*, *Pseudomonas mandelii*, and *Rahnella aquatilis* from *Dendroctonus ponderosae*	*S. marcescens* reduced 49–79% of 3-carene and (−)-β-pinene, and *P. mandelii* decreased concentrations of all monoterpenes by 15–24%, while *R. aquatilis* decreased (−)-α-pinene (38%) and (+)-α-pinene (46%) by 40% and 45% (by GC-MS), respectively	[41]
		*Pseudomonas*, *Rahnella*, *Serratia*, and *Burkholderia* in *D. ponderosae*	Genera contained most genes involved in terpene degradation (by metagenomics)	[42]
		*Serratia* sp., *Pseudomonas* sp., and *Rahnella aquatilis* in *Dendroctonus valens*	Degraded 20–50% of α-pinene (by GC-MS)	[43]
	Diterpene	gut microbiota of *Hylobius abietis*	Gut bacterial community of *H. abietis* reduced most diterpenes, and metagenomic analysis results showed gut community contained 10 degradation genes (*dit*) (by metagenome sequencing and GC-MS)	[44]
	Saponin	*Acinetobacter* sp. in *Curculio chinensis*	*Acinetobacter* sp. in *C. chinensis* enriched after treating with saponin, and when incubating bacteria with saponin for 72 h, saponin content significantly decreased from 4.054 to 1.867 mg/mL (by 16S rRNA metagenome sequencing and HPLC)	[45]
	Azadirachtin	*Serratia marcescens* in *Rhodnius prolixus*	*S. marcescens* load in *R. prolixus* increased when fed diet containing azadirachtin at 1 μg/mL (by qRT-PCR)	[46]
Alkaloid	Caffeine	*Pseudomonas fulva* in *Hypothenemus hampei*	*P. fulva* processed gene coding one subunit of caffeine demethylase, and reinstatement of *P. fulva* in germ-free *H. hampei* degraded all caffeine consumed (by 16S rRNA gene sequencing and GC-MS)	[48]
	Aconitine, nicotine	entire gut bacteria of *Dendrolimus superans* and *Lymantria dispar*	Abundance of genus *Pseudomonas* in *D. superans* larvae increased, but *Serratia* and *Enterobacter* decreased, and *L. dispar* larvae fed on aconitine-treated diet and nicotine-treated diet shared dominant bacteria *Enterococcus* (by 16S rRNA gene sequencing)	[55]
Phenol	Phenolic glycoside	*Erwinia dacicola* in *Bactrocera olea*	Larvae developed in unripe olive harbored more *E. dacicola* (by 16S rRNA gene sequencing)	[49]
	Phenolic naringenin	*Novosphingobium* sp. in *D. valens*	*Novosphingobium* sp. possesses putative genes involved in degradation of naringenin, and *D. valens* supplied with *Novosphingobium* sp. acquired protection against naringenin (by metagenomic analysis)	[50]
	Tannins	*Acinetobacter* sp. in *Lymantria dispar*	Condensed tannins improved growth of *Acinetobacter* sp. by 15% (by measuring the optical density)	[51]
Glucosinolate		*Pantoea* sp. Pc8 in *Psylliodes chrysocephala*	Laboratory-reared and field-collected *P. chrysocephala* all contained three core genera *Pantoea*, *Acinetobacter* and *Pseudomonas*, and reintroduction of *Pantoea* sp. Pc8 in antibiotic-fed beetles restored isothiocyanate degradation ability in vivo (by 16S rRNA gene sequencing and LC-MS)	[52]
Oxalate		*Ishikawaella capsulata* in *Megacopta punctatissima*	Encodes genes of oxalate decarboxylase (by whole-genome shotgun sequencing)	[53]

## Data Availability

Not applicable.

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
