# Peer review of "The Role of Insect Symbiotic Bacteria in Metabolizing Phytochemicals and Agrochemicals"

_insects, 2022, doi:10.3390/insects13070583_

Round 1

Reviewer 1 Report

In this review, Zhao, Guo et al. summarize the research progress on the function of insect microbiota in metabolizing xenobiotics, and further give the perspectives in this research field. Considering that the microbiota-mediated xenobiotics metabolism is a hot topic, this article is timely and valuable. My major concerns are:

The ‘Simple Summary’ did not cover the whole scope, only mentioned the abuse of agrochemicals and the degradation of them by gut microbiota, you should present a more comprehensive Simple Summary.

Across the paper, you mentioned the ‘plant secondary substance’ and ‘plant secondary metabolites’ various times. I think the authors want to express the same meaning, consider replacing them with the same expression.

  1. L28, ‘Give the importance…’ should be written to ‘Given the importance…’
  2. L81-L82, you mentioned ‘For generalist insects, This polyphagous habit’ and I don’t understand what you want to point out here. Please check this sentence.
  3. In the title of the Table.1 and 3.1 (L107 and L108), you mentioned the ‘pesticides’ two times. I think you can use the same word ‘agrochemicals’ as the same as the above description (L93).
  4. L140, the name of gene should be italic.
  5. L166, ‘play critical roles’

Author Response

Dear Editor and reviewers:

Thank you for your concern about our manuscript titled “The role of insect microbiota in metabolizing xenobiotics” (insects-1720027). As you are mentioned, there are several important issues that need to be addressed. According to your nice suggestions, we have made some revisions, and the details are listed below.

Reviewer: 1

Comment 1. The ‘Simple Summary’ did not cover the whole scope, only mentioned the abuse of agrochemicals and the degradation of them by gut microbiota, you should present a more comprehensive Simple Summary.

Reply: Thanks for your suggestion. We have presented a new Simple Summary, please see it in the revised version of this manuscript.

Comment 2. Across the paper, you mentioned the ‘plant secondary substance’ and ‘plant secondary metabolites’ various times. I think the authors want to express the same meaning, consider replacing them with the same expression.

Reply: Thank you for you advice. The ‘plant secondary metabolites’ across the paper has replaced with ‘plant secondary substance’.

Comment 3. L28, ‘Give the importance…’ should be written to ‘Given the importance…’

Reply: Thank you for your kindly suggestion. This word has been corrected.

Comment 4. L81-L82, you mentioned ‘For generalist insects, This polyphagous habit’ and I don’t understand what you want to point out here. Please check this sentence.

Reply: Sorry for the missing words in this sentence. Here what I want to express was that “For generalist insects, to some extent, their polyphagous habits rely on some symbiotic microbes to adapt plant secondary substances from different host plants”.

Comment 5. In the title of the Table.1 and 3.1 (L107 and L108), you mentioned the ‘pesticides’ two times. I think you can use the same word ‘agrochemicals’ as the same as the above description (L93).

Reply: Thanks a lot, the ‘pesticides’ in section 3.1 has changed to ‘agrochemicals’.

Comment 6. L140, the name of gene should be italic.

Reply: Thank you, the gene name has been italicized.

Comment 7. ‘play critical roles’

Reply: Thank you. ‘play critical role’ in part 5 has changed to ‘plays critical roles’.

Reviewer 2 Report

In this review, the authors discuss the role of the microbiota in xenobiotic metabolism. The authors organized this review into three main parts, the first dealing with allelochemical compounds of plants, the second with agrochemicals, and the third with mycotoxins and antibiotics. The topic chosen by the authors is certainly interesting, both from the applicative point of view and from the point of view of basic research. The study is well written and sounds scientifically good. However, some points should be further developed. Indeed, the literature is full of examples that could be cited by the authors. For this reason, I suggest approving the study after some changes.

General: I enjoyed this study, but I believe the literature should be expanded and more images should be included. In general, fungi are hardly mentioned (except when it comes to mycotoxins). Are there examples of other degraded or fungal-produced xenobiotics? Also, the study talks about microbiota. However, many of the examples speak of single bacteria, perhaps the authors could also speak of entire communities (metagenomics, shotgun/metabarcoding)

Comments:

1)    Introduction: First, the authors need to clarify what xenobiotic means. Usually, a xenobiotic is a substance that is not produced by an organism but is found in this organism itself. However, there are broader definitions: a xenobiotic is a substance that should not be found in an organism (In this case, the substances not produced by the insect but derived from the diet are not xenobiotic). According to these two definitions, some plant compounds may be xenobiotic (since they are not produced by the insect) or not (since these compounds are expected to be found in the insect). I believe the authors call xenobiotic any substance that is not produced by the insect in which it was re-encountered. This point should be clarified.

2)    Insect microbiota confer resistance to plant allelochemicals: I believe this chapter needs to be developed better. First, the authors cite a limited number of examples. The bacteria that help insects fight toxic plant compounds are numerous. Secondly, in addition to the simple detoxification action, bacteria can convert toxic compounds into useful compounds that can be used as semiochemicals (social communication, sexual communication, and more). This point should be discussed.

3)    Association between microbiota and insects’ adaptation to agrochemicals: I think the authors could better describe the indirect role of insects by expanding the literature. For example, they could report a table with genes that are regulated by the presence of certain bacteria.

4)    Degradation of mycotoxins and antibiotics by insect microbes: as the comment 2. The authors focus heavily on the degradation of antibiotics or mycotoxins. The authors do not consider other aspects, such as the fact that many antibiotic-producing bacteria have been isolated or described from insects. I know that many insects have a symbiosis with antibiotic-producing bacteria. These symbioses are positive. In this case, I believe the ecological role of antibiotic production needs to be better addressed by including the function these molecules may have and the possibility to isolate antibiotic-producing bacteria from insects (biotechnological application).

5)    Line 82, please rephrase.

Author Response

Dear Editor and reviewers:

Thank you for your concern about our manuscript titled “The role of insect microbiota in metabolizing xenobiotics” (insects-1720027). As you are mentioned, there are several important issues that need to be addressed. According to your nice suggestions, we have made some revisions, and the details are listed below.

Reviewer: 2

Comment 1. I enjoyed this study, but I believe the literature should be expanded and more images should be included. In general, fungi are hardly mentioned (except when it comes to mycotoxins). Are there examples of other degraded or fungal-produced xenobiotics? Also, the study talks about microbiota. However, many of the examples speak of single bacteria, perhaps the authors could also speak of entire communities (metagenomics, shotgun/metabarcoding).

Reply: Thank you for your nice suggestion. After revision, the topic of this paper will focus on “The role of insect symbiotic bacteria in metabolizing plant secondary substances and pesticides”. In the introduction part, the role of fungi and some other microbes were described briefly, “For example, the fungal yeast-like symbiotes in planthoppers and aphids were vital to hosts in essential amino acid synthesis and vitamin supply [10-12]. The polydnavirus from parasitoid wasps could disorder the immune system of host insects to ensure the survival of wasp offspring, and three partiti-like viruses identified from African armyworm (Spodoptera exempta) could enhance the resistance of S. exempta to nucleopolyhedrovirus [13, 14]. For wood-feeding lower termites, they relied on symbiotic flagellates to decompose the lignocelluloses in their plant diet, and methanogenic archaea to produce methane [15, 16]. In terms of symbiotic bacteria, they comprise the most abundant microorganism species in insects and mainly distribute in gut, including Proteobacteria, Bacteroidetes, Firmicutes, Clostridia, Actinomycetes and others, which have been shown to contribute to the development, behavior, communication and adaptation of host insects”. In addition, the publication speak of entire communities were also added in the manuscript (see part 2 and Table 1).

Comment 2. First, the authors need to clarify what xenobiotic means. Usually, a xenobiotic is a substance that is not produced by an organism but is found in this organism itself. However, there are broader definitions: a xenobiotic is a substance that should not be found in an organism (In this case, the substances not produced by the insect but derived from the diet are not xenobiotic). According to these two definitions, some plant compounds may be xenobiotic (since they are not produced by the insect) or not (since these compounds are expected to be found in the insect). I believe the authors call xenobiotic any substance that is not produced by the insect in which it was re-encountered. This point should be clarified.

Reply: Thank you for your kindly explanation of the definitions of xenobiotic. Indeed, the definition of xenobiotic across the paper was too broad and inappropriate, so we would like to focus on plant secondary substances and pesticides. The title of this paper has corrected to ‘The role of insect symbiotic bacteria in metabolizing plant secondary substances and pesticides’. And the ‘xenobiotic’ across the paper was also replacted with plant secondary substances or pesticides.

Comment 3. Insect microbiota confer resistance to plant allelochemicals: I believe this chapter needs to be developed better. First, the authors cite a limited number of examples. The bacteria that help insects fight toxic plant compounds are numerous. Secondly, in addition to the simple detoxification action, bacteria can convert toxic compounds into useful compounds that can be used as semiochemicals (social communication, sexual communication, and more). This point should be discussed.

Reply: Thank you. The part of insect microbiota confer resistance to plant allelochemicals has been revised. More studies have been added in this part and the role of insect bacteria in converting toxic compounds into pheromone compounds has been described (more details please see in section 2).

Comment 4. Association between microbiota and insects’ adaptation to agrochemicals: I think the authors could better describe the indirect role of insects by expanding the literature. For example, they could report a table with genes that are regulated by the presence of certain bacteria.

Reply: Thank you. The part has expanded more studies. And the studies with genes that wereregulated by the presence of certain bacteria have added in table 2 (more details please see in section 3).

Comment 5. Degradation of mycotoxins and antibiotics by insect microbes: as the comment 2. The authors focus heavily on the degradation of antibiotics or mycotoxins. The authors do not consider other aspects, such as the fact that many antibiotic-producing bacteria have been isolated or described from insects. I know that many insects have a symbiosis with antibiotic-producing bacteria. These symbioses are positive. In this case, I believe the ecological role of antibiotic production needs to be better addressed by including the function these molecules may have and the possibility to isolate antibiotic-producing bacteria from insects (biotechnological application).

Reply: Thank you for your nice comments. This part has been revised, and the examples of antibiotic-producing bacteria have been listed. “On the other hand, some insect bacteria could produce antimicrobial compounds which contribute to protection from pathogens. For example, the gut bacterium Enterococcus mundtii in Spodoptera littoralis could secrete an antimicrobial (mundticin KS) against the invading bacteria, and the purified mundticin could cures the larvae infected with E. faecalis [21]. Besides, the cockroaches also carry bacteria that could produce metabolites or proteins with potential industrial applications, such as the antibiotic-producing Streptomyces strain, Bacillus strain, Enterococcus strain and Pseudomonas species, all of which may be suitable for development as pharmaceuticals or plant protection products and provide opportunities for biotechnological application”

Comment 6. Line 82, please rephrase.

Reply: Thank you. Here has been revised.

Reviewer 3 Report

The extremely short mini-review by Zhao et al gives a short summary about cases of insect-symbionts that are known for the detoxification of pesticides or other metabolites, which might be harmful for the insect.
The title is a bit misleading as phytochemicals, antibiotics and mycotoxins are not really xenobiotics. 
Overall, the English needs to be revised carefully as many sentences make grammatically no sense at all.
Despite the short length this MS appears little focused and starts to introduce each metabolite groups with a broad introduction (e.g. line 70-78, line 94-101, line 144-152). These should emphasize the general importance for humans and agriculture but distract more from the main topic by losing focus.
What is sad that a real discussion is lacking. This manuscript merely lists several cases without further discussing their biological relevance. It might be easy to isolate a bacterium from the gut of an insect and show in vitro that it degrades a certain metabolite, but does this also mean that the insect depends/benefits from these bacteria? It makes a big difference if a study shows only that bacterium x can degrade metabolite Y (evidence for biological importance), or if a study really demonstrates under natural conditions that the host insect has a disadvantage when lacking their symbiont.  
Fig 1 From all the great examples of insect with important symbionts, why has a caterpillar been taken for this picture as example? While caterpillars are least dependent on microbes compared many other insects and maybe the worst example for this, as they perform well under antibiotic exposure.
Table 1 What should "degrada" mean? The narrow typesetting of the columns with overly large font make this table difficult to read e.g. does Acinetobacter lwoffii belong to Blatta or Drosophila?

Specific comments:
line 17 The abstract mentions fungi, virus, protozoa, and archaea, which play no role in this manuscript.
line 27 Definition of xenobiotics too broad.
line 57 "which always influences by food resources" awkward sentence, needs to be revised.
line 66 The English grammar needs to be revised.
line 117 Replace "the microbiota" with "the bacterium"
line 157-158 The sentence needs to be corrected grammatically
line 167 Mycotoxins, antibiotics and plant metabolites are no xenobiotics.
line 169-170 grammar

Author Response

Dear Editor and reviewers:

Thank you for your concern about our manuscript titled “The role of insect microbiota in metabolizing xenobiotics” (insects-1720027). As you are mentioned, there are several important issues that need to be addressed. According to your nice suggestions, we have made some revisions, and the details are listed below.

Reviewer: 3

Comment 1. The title is a bit misleading as phytochemicals, antibiotics and mycotoxins are not really xenobiotics. 

Reply: Thank you for your kindly advice. The xenobiotic in the title was a bit misleading and the definition of xenobiotic across the paper was not very correct, so the title of this paper has corrected to ‘The role of insect symbiotic bacteria in metabolizing plant secondary substances and pesticides’. And the ‘xenobiotic’ across the paper was also replacted with plant secondary substances or pesticides.

Comment 2. Overall, the English needs to be revised carefully as many sentences make grammatically no sense at all.

Reply: Sorry for the grammar issue in this paper. We have used the English editing services Liwen Bianji (Edanz) (https://www.liwenbianji.cn) for editing the language of a draft of this manuscript.

Comment 3. Despite the short length this MS appears little focused and starts to introduce each metabolite groups with a broad introduction (e.g. line 70-78, line 94-101, line 144-152). These should emphasize the general importance for humans and agriculture but distract more from the main topic by losing focus.

Reply: Thank you for your kindly suggestion. After revising, the topic of this manuscript mainly focuses on the function of insect bacteria in metabolizing plant secondary substances and pesticides. And the importance of the plant secondary substances and pesticides for humans and agriculture has been introduced.

Comment 4. What is sad that a real discussion is lacking. This manuscript merely lists several cases without further discussing their biological relevance. It might be easy to isolate a bacterium from the gut of an insect and show in vitro that it degrades a certain metabolite, but does this also mean that the insect depends/benefits from these bacteria? It makes a big difference if a study shows only that bacterium x can degrade metabolite Y (evidence for biological importance), or if a study really demonstrates under natural conditions that the host insect has a disadvantage when lacking their symbiont.  

Reply: According to your nice comments, the role of insect bacterium verified in vivo and vitro have been described and discussed in the manuscript. For some bacteria, the eliminating of them in host insects will decrease their adaptation to plant secondary substances and pesticides. After supplying germ-free hosts with the degrading bacteria, the resistance of hosts to detrimental substances will restore.

Comment 5. Fig 1 From all the great examples of insect with important symbionts, why has a caterpillar been taken for this picture as example? While caterpillars are least dependent on microbes compared many other insects and maybe the worst example for this, as they perform well under antibiotic exposure.
Table 1 What should "degrada" mean? The narrow typesetting of the columns with overly large font make this table difficult to read e.g. does Acinetobacter lwoffii belong to Blatta or Drosophila?

Reply: Thank you for your nice suggestion. Caterpillars used in figure 1 since it was one of my research insects. According to my research results, if we eliminate the symbiotic bacteria of caterpillars at egg stage and then rear the hatching first instar larvae in germ-free conditions, the development and survival of caterpillars were significantly reduced, and most of them cannot develop into third instar (unpublished data). If caterpillars perform well under antibiotic exposure, it may because the bacteria were not eliminated completely, since play important roles in many aspects of the biology and physiology of insect. The elimination of gut microbes can be verified by plating the homogenate of antibiotic-treated insects on LB and PDA agar plates and performed PCR using bacterial primers and fungal primers. The "degrada" should be “degrade”. The table 2 has been revised, and Acinetobacter lwoffii belong to Blatta.

Comment 6. line 17 The abstract mentions fungi, virus, protozoa, and archaea, which play no role in this manuscript.

Reply: Thank you for your reminding. Of course, apart from bacterium, the other microbes also play important roles in insects. In the introduction part, the role of fungi and some other microbes were described briefly, “For example, the fungal yeast-like symbiotes in planthoppers and aphids were vital to hosts in essential amino acid synthesis and vitamin supply [10-12]. The polydnavirus from parasitoid wasps could disorder the immune system of host insects to ensure the survival of wasp offspring, and three partiti-like viruses identified from African armyworm (Spodoptera exempta) could enhance the resistance of S. exempta to nucleopolyhedrovirus [13, 14]. For wood-feeding lower termites, they relied on symbiotic flagellates to decompose the lignocelluloses in their plant diet, and methanogenic archaea to produce methane [15, 16]. In terms of symbiotic bacteria, they comprise the most abundant microorganism species in insects and mainly distribute in gut, including Proteobacteria, Bacteroidetes, Firmicutes, Clostridia, Actinomycetes and others, which have been shown to contribute to the development, behavior, communication and adaptation of host insects”.

Comment 7. line 27 Definition of xenobiotics too broad.

Reply: Yes, we have deleted the ‘xenobiotic’ across the paper, and replacted it with plant secondary substances or pesticides.

Comment 8. line 57 "which always influences by food resources" awkward sentence, needs to be revised.

Reply: Thank you. This sentence has been revised to “In addition, the composition of bacterial communities in insects will be influenced by food resources, environmental factors, pathogenic microbials or the exposed detrimental substances”.

Comment 9. line 66 The English grammar needs to be revised.

Reply: Thank you. Here the English grammar has been checked and revised.

Comment 10. line 117 Replace "the microbiota" with "the bacterium"

Reply: Thank you. Here "the microbiota" has replaced with "the bacterium"

Comment 11. line 157-158 The sentence needs to be corrected grammatically

Reply: Yes, here the grammar has been corrected.

Comment 12. line 167 Mycotoxins, antibiotics and plant metabolites are no xenobiotics.

Reply: Thank you. The word ‘xenobiotic’ has been deleted from the revised manuscript since the definition mentioned in previous version was not correct.

Comment 13. line 169-170 grammar

Reply: Yes, here the grammar has been checked and corrected.

Round 2

Reviewer 2 Report

The authors responded comprehensively to all comments. I recommend publishing the study in its present form.

Author Response

Thank you again for your nice suggestion

Reviewer 3 Report

The manuscript has been extended and largely improved and authors have implemented several suggestions by the reviewers. But following points need to be adressed.
Title: pesticides -> agrochemicals
line 28: chemicals -> agrochemicals
I think to make it consistent the word 'pesticide' or 'chemicals' should be replaced with 'agrochemicals' within the entire manuscript. The authors should also decide if they want to use 'plant secondary substances' 'allelochemicals' or 'phytochemicals' within the manuscript, but not switching between these terms. Fig 1 has 'phytochemicals' the title mentions 'plant secondary substances' and the section headings are about 'allelochemicals'. This is not only inconsistent, but very confusing to other readers. Simply stick to one term and use it consistently throughout the text.

Despite the use of an external english proofreading sevice, there are still several issues with the grammar that need to be fixed. Here are a few examples for spell check requirements (though it is not my job to find these errors):
line 72: studied
line 78: This sentences has not been revised gramatically ('is prospected' or 'will be prospected')
line 93: remove 'a'
line 106: bacteria
line 119: 'a research' -> 'a study'
line 143-144: As another important plant allelochemical[s], phenols could inhibit[e] herbivorous [insects] by inducing reactive oxygen production.
line 174: 'an ideal choice' 'for microbe-based'
line 183: express
line 184: have been developed
line 188: 'but recently insect bacterium were also reported' ??? -> 'but recently insect-associated bacteria have also been reported'
line 198: 'strains' 'receive'
line 201: dengue
line 212: 'increase to' 'CFUs'
line 222: 'Genome'
line 228: 'Genome'
line 291: 'of of'

Author Response

Dear Editor and reviewers:

Thank you for your concern about our manuscript titled“The role of insect symbiotic bacteria in metabolizing phytochemicals and agrochemicals”(insects-1720027). As you are mentioned, there are several important issues that need to be addressed. According to your nice suggestions, we have made some revisions, and the details are listed below.

Reviewer: 3

Comment 1. In the title, the word 'pesticides' should be written to 'agrochemicals'.

Reply: Thank you for your kindly suggestion. This word has been corrected.

Comment 2. Line 28, the word 'chemicals' should be written to 'agrochemicals'.

Reply: Thank you for your kindly advice. This word has been corrected.

Comment 3. The word 'pesticide' or 'chemicals' should be replaced with 'agrochemicals' within the entire manuscript.

Reply: Thanks for your suggestion. The word 'pesticide' or 'chemicals' across the paper has replaced with 'agrochemicals'.

Comment 4. The authors should also decide if they want to use 'plant secondary substances' 'allelochemicals' or 'phytochemicals' within the manuscript, but not switching between these terms. Fig 1 has 'phytochemicals' the title mentions 'plant secondary substances' and the section headings are about 'allelochemicals'.

Reply: Thank you for you advice. The words 'plant secondary substances' 'allelochemicals' or 'phytochemicals' has replaced with  'phytochemicals' in the entire manuscript.

Comment 5. Line 72, the word 'studies' should be written to 'studied'.

Reply: Thank you for your kindly suggestion. This word has been corrected.

Comment 6. Line 78: This sentences has not been revised gramatically ('is prospected' or 'will be prospected').

Reply: Thanks a lot, We haves been revised gramatically in this sentences.

Comment 7. Line 93: This sentences should be remove 'a'.

Reply: Thank you, We haves been removed 'a' in this sentences.

Comment 8. Line 106: the word 'bacterium' should be written to 'bacteria'.

Reply: Thanks a lot, This word has been corrected.

Comment 9. Line 119: the word 'a research' should be written to 'a study'.

Reply: Reply: Thank you for your kindly advice. This word has been corrected.

Comment 10. Line 143-144: This sentence 'As another important plant allelochemical[s], phenols could inhibit[e] herbivorous [insects] by inducing reactive oxygen production' should be written to 'As another important plant allelochemicals, phenols could inhibite herbivorous insects by inducing reactive oxygen production'.

Reply: Thank you for your kindly suggestion. This sentence has been corrected.

Comment 11. Line 174: the 'an ideal choice' 'for the  microbe-based' should be written to 'an ideal choice' 'for microbe-based'.

Reply: Thank you for you advice, this sentence has been corrected.

Comment 12. Line 183: the word 'expressing' should be written to ' express'.

Reply: Thanks you. This word has been corrected.

Comment 13. Line 184: the 'has developed' should be written to ' have been developed'.

Reply:Thanks a lot. this phrase has been corrected.

Comment 14. Line 188: the 'but recently insect bacterium were also reported' ??? should be written to 'but recently insect-associated bacteria have also been reported'.

Reply: Thank you. This sentence has been corrected.

Comment 15. Line 198: the word 'strain and received' should be written to ' strains and 'receive'.

Reply: Thank you. This word has been corrected.

Comment 16. Line 201: the word 'denzue' should be written to ' dengue'.

Reply: Thank a lot. This word has been corrected.

Comment 17. Line 212: this sentence  'the bacterial community would increase 107 to 108 CFU' should be written to ' the bacterial community would increase to 107 to 108 CFUs'.

Reply: Thank you for your kindly suggestion. This sentence has been corrected.

Comment 18. Line 222: the word 'Gemone' should be written to ' Genome'.

Reply: Thank you. This word has been corrected.

Comment 19. Line 228: the word 'Gemone' should be written to ' Genome'.

Reply: Thank you. This word has been corrected.

Comment 20. Line 291: this sentence should be removed a 'of'.

Reply: Thank a lot. This word has been removed.